# Reduction of Ammonia Emissions from Laying Hen Manure in a Closed Composting Process Using Gas-Permeable Membrane Technology

María Soto-Herranz [1,*] , Mercedes Sánchez-Báscones [1] , Juan Manuel Antolín-Rodríguez [1] and Pablo Martín-Ramos [2]

1   Departamento de Ciencias Agroforestales, ETSIIAA, Universidad de Valladolid, Avenida de Madrid 44, 34004 Palencia, Spain; mercedes.sanchez@uva.es (M.S.-B.); juanmanuel.antolin@uva.es (J.M.A.-R.)
2   Department of Agricultural and Environmental Sciences, EPS, Instituto Universitario de Investigación en Ciencias Ambientales de Aragón (IUCA), Universidad de Zaragoza, Carretera de Cuarte, s/n, 22071 Huesca, Spain; pmr@unizar.es
*   Correspondence: maria.soto.herranz@alumnos.uva.es

**Abstract:** Nitrogen losses during composting processes lead to emissions problems and reduce the compost fertilizer value. Gas-permeable membranes (GPM) are a promising approach to address the challenge of reducing nitrogen losses in composting processes. This study investigated the applicability of two GPM membrane systems to recover N released during the closed composting process of laying hen manure. The ammonia ($NH_3$) capture process was performed using two different systems over a period of 44 days: the first system (S1) consisted of 120 m of an expanded polytetrafluoroethylene (ePTFE) membrane installed inside a 3.7 $m^3$ portable, closed aerobic composter with forced ventilation; the second system (S2) consisted of 474 m of an ePTFE membrane placed inside as an external module designed for $NH_3$ capture, connected to a closed aerobic composter through a pipe. In both cases, a 1 N $H_2SO_4$ acidic $NH_3$ capture solution was circulated inside the membranes at a flow rate of 2.1 L·h$^{-1}$. The amount of total ammonia nitrogen (TAN) recovered was similar in the two systems (0.61 kg in S1 and 0.65 kg in S2) due to the chosen membrane surface areas, but the TAN recovery rate was six times higher in system S1 (6.9 g TAN·m$^{-2}$·day$^{-1}$) than in system S2 (1.9 g TAN·m$^{-2}$·day$^{-1}$) due to the presence of a higher $NH_3$ concentration in the air in contact with the membrane. Given that the $NH_3$ concentration in the atmosphere of the membrane compartment directly influences the $NH_3$ capture, better performance of the GPM recovery system may be attained by installing it directly inside the closed aerobic composters. Regardless of the chosen configuration, this technology allows N recovery as a stable and concentrated 1.4% N ammonium salt solution, which can be used for fertigation. The presented GPM systems may be used in community composting systems with low volumes of waste to be treated or in livestock facilities that have implemented best available techniques such as solid–liquid separation or anaerobic digestion, provided that the use of GPM technology in combination with these techniques also contributes to odor mitigation and improves biogas yields.

**Keywords:** poultry manure; gas-permeable membrane; ammonia recovery

## 1. Introduction

In the European Union (EU), the livestock production sector generates around 1400 million tons of manure annually, out of which only 7.8% (equivalent to 108 million tons per year, containing 556,000 tons of nitrogen and 139,000 tons of phosphorus) is processed. The highest manure processing percentages correspond to Italy, Greece, and Germany, accounting for 36.8, 34.6, and 14.8% of their manure production, respectively [1].

Regarding poultry manure production, there has been an increase during the last 20 years as a result of the considerable growth of the poultry industry in most EU member

countries [2]. At the EU level, the census of laying hens from 2013 to the present has ranged from 380.4 to 413.2 million heads; in Spain, its evolution has been stable over the last 10 years, with values close to 50 million heads [3]. This livestock activity generates a large volume of waste, whose estimated production (in thousands of tons) in the EU and Spain is 109,518 and 12,726 tons, respectively [1]. As production is concentrated in specific areas, the management of the generated manure is a challenge [4], with important environmental and health implications.

Environmental problems resulting from the ever-increasing production of organic waste include soil acidification, nitrates in groundwater, eutrophication of water [5], odors, and gaseous emissions [6] (if not properly processed). In particular, poultry waste generates large amounts of ammonia ($NH_3$) emissions, produced due to microbial degradation of uric acid present in manure [7,8]. Ammonia emissions in Spain have remained relatively stable since 1990, peaking in the first half of the 2000s. In recent years, they have increased again due to a combination of increased livestock numbers and the use of inorganic fertilizers. In 2017, six member states (Austria, Croatia, Germany, Ireland, the Netherlands, and Spain) exceeded their $NH_3$ ceilings, with the highest exceedances in Spain (47%) and Croatia (25%). The largest emitter of $NH_3$ in the EU is Germany, followed by France and Spain. In 2019, the last full year for which data are available, Spain reached emissions of 471 kt [9]. Directive EU/2016/2284 [10] establishes that $NH_3$ emissions in Spain must be reduced by −3% in the 2020–2029 period (compared to 2005) and that global emissions must be reduced by −16% (compared to 2005 emissions) from 2030 onwards, taking into account the emission ceiling of 353 kt. As livestock and agriculture are the primary sources of $NH_3$ emissions (accounting for over 95% of the total), reductions should be concentrated in these sectors. The projection of emissions in the baseline scenario (WeM), taking only into account existing measures, foresees noncompliance with emission reduction commitments for most of the projection period. However, in the "with additional measures" (WAM) scenario, the emission ceilings set for the period 2020–2030 would be met [11]. For European countries, almost one-third of all EU member states reported projections above their respective non-methane volatile organic compounds (NMVOCs) reduction commitments for 2020. Looking ahead to 2030, EU member states need to make further efforts to meet their emission reduction commitments, as more than half are not on track to meet their agreed reduction commitments for $NH_3$, NMVOCs, NOx, and PM2.5 [9].

In addition to the environmental problems, N losses in the composition of the waste must also be considered. Furthermore, prolonged exposure to N can affect humans and animals' respiratory and cardiovascular systems [12] and can cause the formation of fine PM2.5 particles in the presence of NOx or SOx [13,14].

Concern about the issues mentioned above has generated a growing demand for technologies to improve this waste management [15]. Among them, composting stands out as a simple, cost-effective, and viable technology for the valorization of these wastes, allowing their hygienic transformation into a homogeneous and stable material with high fertilizer value. Microorganisms decompose the organic matter in the waste under aerobic conditions and in environments with certain humidity and temperature, using N and C to produce their own biomass [16], while reducing the survival of pathogenic bacteria and reducing the odor of the waste and the volume generated [17,18]. In this regard, it is worth noting that, although poultry manure is widely used as a fertilizer in agriculture due to its high nutrient content [17], it is a major source of pathogenic microorganisms, such as *Salmonella* spp., *Escherichia coli*, *Staphylococcus* spp., *Streptococcus* spp., *Clostridium* spp., *Listeria* spp., *Campylobacter* spp., *Corynebacterium* spp., and *Mycobacterium* spp. [19]. The application of untreated chicken manure may cause environmental problems, release of phytotoxic substances, human diseases, or food-borne outbreaks. Therefore, it should be appropriately treated and managed before application as a biofertilizer [20], and composting is a suitable manure management modality for its treatment.

Despite the aforementioned advantages, if the factors for correct composting are not adequately controlled, the process can result in significant emissions into the air, which

reduce the agronomic value of the compost and damage the environment through the release of gases such as $NH_3$, $N_2O$, and $CH_4$. In particular, $NH_3$ emissions account for 46.8–94% of the total N losses during composting [21,22]. Therefore, N loss due to $NH_3$ emissions would be the main reason behind the reduction of compost quality and the generation of odor pollution in the composting process [23–25].

Factors affecting $NH_3$ emissions during composting include temperature, moisture content, pH, initial N content of the substrate, aeration, aeration rates, and the type of composting process [7]. $NH_3$ emissions also depend on the phase of the composting process (the first phase comprises the decomposition of the biodegradable material by microbial activity and stabilization of the organic residue [26], while the second phase consists of transforming a part of the organic material remaining as humic substances [26,27]). Pagans et al., Zhou et al., and Awasthi et al. [28–30] have shown that $NH_3$ emissions increase exponentially during the first thermophilic stage (>45 °C) and linearly within the final mesophilic stage (25–40 °C) within the first phase of the process. Therefore, most of the $NH_3$ emissions occur in the first three weeks of the composting period [31]

To reduce N losses associated with these emissions, the control of parameters such as pH [32,33] and the C/N ratio [34,35], the use of adsorption materials and chemicals [36–38], and the use of microorganisms [39] have been explored. Several studies have shown that the application of additives such as biochar [40,41], $MgCl_2$ and $FeSO_4$ [42], Mg (OH)c and $H_3PO_4$ [43], clay [44], or bacteria inoculated into sludge and mushroom residues [39] are effective strategies to decrease $NH_3$ emissions during the composting process. However, although physicochemical additives have good N retention properties, they have the disadvantage of contributing saline ions and creating an unknown accumulation effect in the soil. Microbial additives are advantageous in terms of cost and environmental friendliness but have not been widely studied to date [37].

There are other technologies that involve the capture of $NH_3$ by neutralization, adsorption, or precipitation, such as reverse osmosis [45], stripping towers [46], adsorption by zeolites [47], phosphate and magnesium precipitation [48], bioadsorbents [49], or gas-permeable membranes (GPM) [50] for recovery and reuse. These technologies do not improve the quality of the compost products but reduce $NH_3$ emissions to the atmosphere and recover N, which is important for the agricultural sector due to the high cost of commercial ammonia fertilizers [51]. Traditional processes have operational limitations and high costs, making their actual application difficult: reverse osmosis requires high pressures; air extraction towers and zeolite adsorption techniques require manure pretreatment; struvite precipitation requires additives [50,52], and the reuse of ammonium-loaded bioadsorbents as biofertilizers or as biocompost is not yet sufficiently investigated [49,53]. On the other hand, GPM technology has low energy consumption (0.18 kWh·kg $NH_3^{-1}$), requires low working pressure, does not require effluent pretreatment, does not require the addition of any alkaline reagents [54], and does not impair livestock operation. At present, the major drawback of this technology is the cost of the membranes.

Gas-permeable membranes can be manufactured in various materials, such as polyethylene, polypropylene, polyvinyl chloride, polyvinylidene fluoride, fluorinated ethylene propylene, perfluoroalkali, ethylenetetrafluoroethylene, polyetheretheretherketone, polytetrafluoroethylene, and expanded polytetrafluoroethylene (ePTFE). The latter material is microporous, flexible, and hydrophobic and is of particular interest due to its high permeability to low-pressure gas flows and its high thermal, mechanical, and chemical resistance [55]. Regardless of the material used, the gas separation process involves the flow of $NH_3$ through the membrane by diffusion, followed by the capture of $NH_3$ in a receiver solution that circulates inside the membrane, forming nonvolatile ammonium salt.

Authors such as Sun et al. and Ma et al. [56,57] have demonstrated a reduction in $NH_3$ and $CH_4$ emissions of 20–30% and 40%, respectively, in aerobic composting reactors using lab-scale ePTFE membrane covers. In addition, Sun et al. [56] also studied an ePTFE semipermeable membrane cover system applied to a farm-scale composting process, achieving a 65% reduction in $NH_3$ emissions. Even though no recovery systems

were implemented in those studies, their promising results have encouraged the research presented herein.

This study aimed to investigate the suitability of installing ePTFE GPM technology in aerobic composting reactors to capture $NH_3$ during the poultry manure composting process, recovering N in the form of a stable and concentrated product, viz. $(NH_4)_2SO_4$. For this purpose, the performance of two configurations in terms of the $NH_3$ recovery efficiency was compared, depending on whether the GPM system was directly installed inside the closed aerobic composting reactor (system 1) or in a separate compartment connected to the closed aerobic composter (system 2).

## 2. Materials and Methods

### 2.1. First System (S1): ePTFE Membranes Installed Inside the Portable Closed Aerobic Composter

The diagram of system S1 is shown in Figure 1, in which a 3.7 $m^3$ portable rectangular aerobic composter was used. Oxygen was supplied to the mixture by forced aeration through pipes inside the aerobic composter, maintaining an aerobic state in the medium. Aeration was applied for 10 min every 24 h, using a 1.5 kW centrifugal fan with a ventilation flow rate of 0.35 $m^3 \cdot min^{-1}$. Humidity control was carried out manually, depending on temperature variations and the moisture content of the mixture.

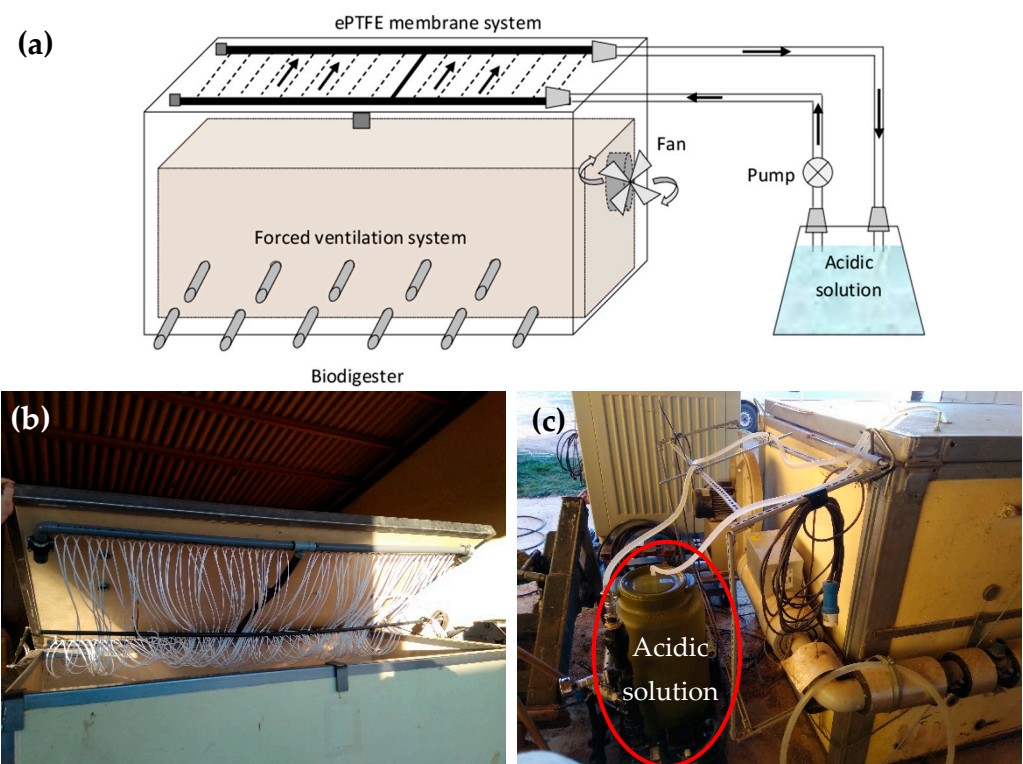

**Figure 1.** System S1: (**a**) schematic of the gas-permeable membrane system for $NH_3$ capture installed inside the closed aerobic composter; (**b**) pilot-scale closed aerobic composter with ePTFE membrane installed in the inner face of the lid; (**c**) acidic solution tank (highlighted in red).

The ePTFE membrane (ZEUS Industrial Products Inc., Orangeburg, SC, USA) system was arranged along the inner face of the aerobic composter lid, parallel to the aeration tubes. Inside the membranes, 1 N $H_2SO_4$ was circulated from an external tank so that the $NH_3$ released during the composting process was captured in the acidic solution in the form of nonvolatile $(NH_4)_2SO_4$ and returned to the tank. The acidic solution circulation rate was 2.1 $L \cdot h^{-1}$. The type of capture solution and flow rate were chosen according to a previous study by our group [55]. The volume of acidic solution for $NH_3$ capture was 45 L. The acidic solution distribution system consisted of two 4 cm diameter PVC pipes

(in black in Figure 1a), forming an inlet and an outlet channel, blind at one end and open at the other. The membranes were placed between the two channels, with a separation of 2.5 cm between them. The inlet and outlet channels were slightly inclined in favor of the direction of the liquid in order to achieve a homogeneous flow through all the membranes once the inlet channel was full.

### 2.2. Second System (S2): ePTFE Membranes Placed in a Compartment Outside the Portable Closed Aerobic Composter

The diagram of system S2 is shown in Figure 2. In this case, a closed aerobic composter with identical characteristics to the one explained in the previous section was connected to an external device developed as part of the LIFE+ AMMONIA TRAPPING project, inside which the ePTFE tubular membranes were placed. This device consists of several elements: frame structure, $NH_3$ absorption receptacle, acidic solution recirculation system, and electrical panel.

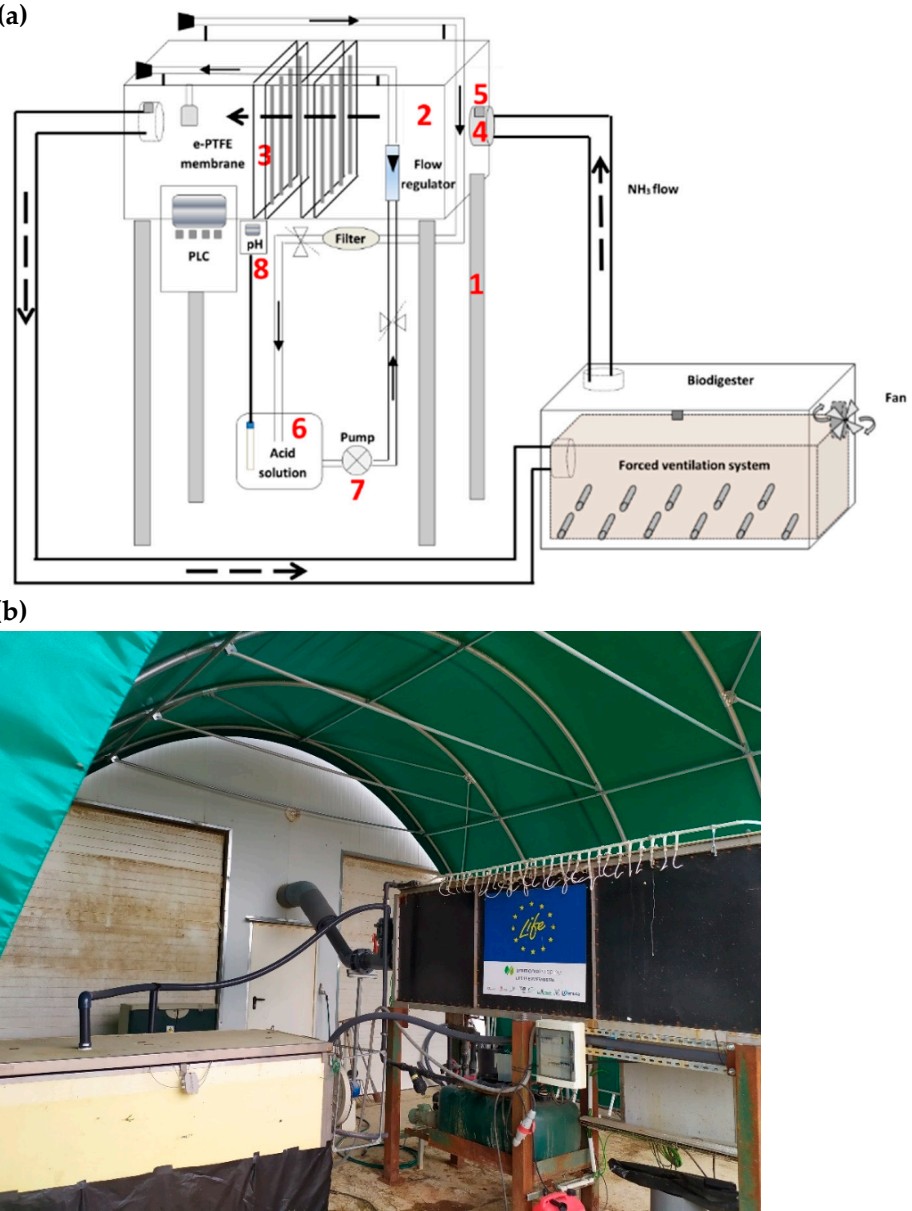

**Figure 2.** System S2: (**a**) schematic of the gas-permeable membrane system installed in an external compartment; (**b**) pilot-scale closed aerobic composter connected to system S2.

The structure (#1 in Figure 2) is made of carbon steel sections coated with corrosion-resistant paint. It serves the function of raising the $NH_3$ collection receptacle off the ground to avoid solid particles and contact with animals. The $NH_3$ absorption receptacle (#2 in the diagram) consists of a structure, frames, and rails made of AISI 304 stainless steel, which support frames for the membranes arranged on a grid. The receptacle structure is covered with high-density polyethene panels. The membranes (#3), vertically distributed over the grids of each of the frames, have an inlet branch and an outlet branch through which the acidic solution circulates, connected to the general recirculation system through flexible joints. In addition, the system consists of a fan (#4) (single-phase, 230 V, 50 Hz, 150 W) and $NH_3$ sensors (#5; model DURTOX IP65-v07; Duran® Electronic, Madrid, Spain). The acidic solution recirculation system consists of a storage tank (#6), a pumping system (#7), and a pH measurement and control system (#8). The pump is of the horizontal centrifugal type, suitable for pumping chemicals. It has a single-phase power supply (230 V, 50 Hz) with a maximum power of 750 W. The acidic solution flow rate was 2.1 L·h$^{-1}$, as in system S1. The volume of acidic solution used for $NH_3$ capture was 150 L.

This device was connected to the aerobic composter through a flexible pipe so that the air present in the atmosphere of the closed aerobic composter was directed toward the external compartment with the membranes and passed through the membranes for cleaning, returning to the closed aerobic composter (Figure 2).

The characteristics of the ePTFE membranes used in the experiments are summarized in Table 1.

**Table 1.** Characteristics of the expanded polytetrafluoroethylene (ePTFE) membranes used in each system.

| Characteristics | System 1 (S1) | System 2 (S2) |
|:---:|:---:|:---:|
| Length (m) | 120 | 474 |
| Absorption surface (m$^2$) | 1.96 | 7.74 |
| Internal diameter (mm) | 4.56 | 4.56 |
| Wall thickness (mm) | 0.64 | 0.64 |
| Density of polymer (g/cm$^3$) | 0.95 | 0.95 |
| Porosity (%) | <60 | <60 |
| Average pore size length (μm) | $12.7 \pm 5.9$ | $12.7 \pm 5.9$ |
| Average pore size width (μm) | $1.3 \pm 0.9$ | $1.3 \pm 0.9$ |

### 2.3. Experimental Conditions

The field experiments were carried out at the poultry farm "La Cañada," located in the municipality of Aldealafuente, province of Soria, Spain (41°40′20.0″ N, 2°19′32.2″ W, 1009 m a.s.l.). The farm has 2250 places for free-range laying hens in three sheds (shed A: 6600 places, shed B: 7100 places, and shed C: 8350 places). Between January and February 2021, the hens and manure were emptied to sanitize the houses before they were filled again. Therefore, manure was generated continuously from March 2021 onwards in each house. For these trials, manure from house C was used.

The structuring agent used was chopped straw (5 mm), obtained from the 2020 wheat harvest. The volume of the mixture (poultry manure + straw) used in the experiment was 2.2 m$^3$, and the poultry manure:straw ratio was 3.5:1 *w/w*.

The experiment lasted 44 days, and three samples of acidic solution were collected on a weekly basis for analysis of pH, electrical conductivity (EC), and ammonia concentration ($NH_4$-N). In addition, representative samples of the mixture were also collected once a week. However, a thorough characterization of the samples was carried out only at the beginning and end of the 44-day period, determining their physical and chemical properties.

### 2.4. Physicochemical Analyses

The product mixture samples at the beginning and the end of the experiment (i.e., after 44 days) were analyzed for moisture content (MC), which was determined by drying

the mixture at 105 °C to constant weight. Volatile solids (VS) were measured by placing dry samples in a muffle furnace at 575 ± 25 °C for four hours. pH and electrical conductivity (EC) were measured with a Crison GLP22 pH-meter (Crison Instruments S.A., Barcelona, Spain). Elemental analysis of carbon and total nitrogen was performed by dry combustion using a LECO CNS928 analyzer (Leco Corp., St. Joseph, MI, USA). Total ammonia nitrogen ($NH_4$-N or TAN) analysis was performed with a Skalar TOC/TN analyzer (Skalar Analytical B.V., Breda, the Netherlands). Total phosphorus was determined by colorimetry in microwave predigested extracts at 430 nm [58]. Assimilable/soluble phosphorus (i.e., "plant-assimilable phosphorus" or "available inorganic phosphorus," which can include small amounts of organic phosphorus, as well as orthophosphate, the form taken up by plants) was extracted in 0.5 N $NaHCO_3$ and subsequently measured by colorimetry at 882 nm [59]. Total nutrient and trace element content was determined by inductively coupled plasma optical emission spectroscopy (ICP-OES) with a Spectro Arcos (Ametek, Kleve, Germany) on 0.3 g of the sample after microwave-assisted digestion with 5 mL 65% $HNO_3$ + 3 mL 37% $H_2O_2$. From the acid-dissolved samples, the TAN concentration was analyzed by distillation (with a Kjeltec$^{TM}$ 8100 nitrogen distillation unit; Foss Iberia S.A., Barcelona, Spain) through distillate capture in borate buffer and subsequent titration with 0.2 mol·L$^{-1}$ HCl [60].

The mixture's temperature was monitored using four probes connected to a data logger (HOBO U12–008; Onset, Bourne, MA, USA) for data storage. To study the influence of the presence of the membrane on $NH_3$ emissions, the concentration of $NH_3$ (g) present in the air chamber of the portable closed aerobic composter atmosphere was measured using a colorimetric tube (Gastec 3La/3M, Japan; error range: ±10%).

Samples of product mixtures (initial control, final control, initial S1, final S1, initial S2, and final S2) were dried for 48 h at 105 °C in an oven and ground to powder using a ball mill. Subsequently, a small sample of the ground powder was taken with a spatula and placed in the cell for analysis by Fourier-transform infrared spectroscopy (FTIR) in a PerkinElmer equipment (Spectrum 400 model) equipped with an attenuated total reflectance (ATR) system (ZnSe prism). The spectra were collected with a 1 cm$^{-1}$ spectral resolution over the 400–4000 cm$^{-1}$ range, taking the interferograms that resulted from co-adding 64 scans. The spectra were then corrected using the advanced ATR correction algorithm available in PerkinElmer Spectrum software.

### 2.5. Calculations

The recovered TAN (expressed in kg) was estimated by the concentration of N captured at the end of the experiment in the acidic solution considering the final volume of the solution.

The N flux across the membrane (TAN recovery rate, expressed in g TAN·m$^{-2}$·day$^{-1}$), which occurs as a consequence of the gas concentration gradient across the membrane [61,62], was determined by considering the TAN recovered per day and the GPM surface area of each system.

The emission rate was calculated using Equation (1) [63]:

$$F = \rho \frac{V}{A} \frac{P}{P_0} \frac{T_0}{T} \frac{dC_t}{dt} \tag{1}$$

where $F$ is the measured gas emission flux (g·m$^{-2}$·day$^{-1}$); $V$ is the volume of the portable closed aerobic composter (m$^3$); $A$ is the area of the portable closed aerobic composter (m$^2$); $dC_t/dt$ is the rate of change of gas concentration in the chamber (ppm·day$^{-1}$); $\rho$ is the gas density at standard conditions (g·m$^{-3}$); $T_0$ is the thermodynamic temperature in standard conditions (273 K); $T$ is the thermodynamic temperature when sampled (in K); $P_0$ is the absolute atmospheric pressure in standard conditions (101 kPa); and $P$ is the pressure at the sampling site (in kPa).

## 3. Results and Discussion

### 3.1. Changes in the Fundamental Physicochemical Parameters during the Composting Process

The daily average temperature of the composting mixture during aerobic composting in S1 and S2 is shown in Figure 3. In both cases, at the beginning of the composting process, the mixture's temperature rapidly increased (up to 68 and 62 °C in S1 and S2, respectively) within 3–4 days and remained between 40 and 60 °C almost until the end of the process. Forced ventilation provided oxygen for the bacteria to decompose the organic matter, increasing the temperature. Subsequently, the temperature gradually decreased to values between 30 and 40 °C. The temperature drop after the third day is attributed to a programmed low aeration frequency (10 min/24 h), which should probably have been more frequent at the beginning of the process to maintain good bacterial activity. The peaks observed during the temperature decrease are due to the recovery of the mixture temperature after a period of ventilation. The temperature variation during the composting process was adequate for uniform and efficient mixture decomposition and was consistent with that reported by other authors [56,64,65].

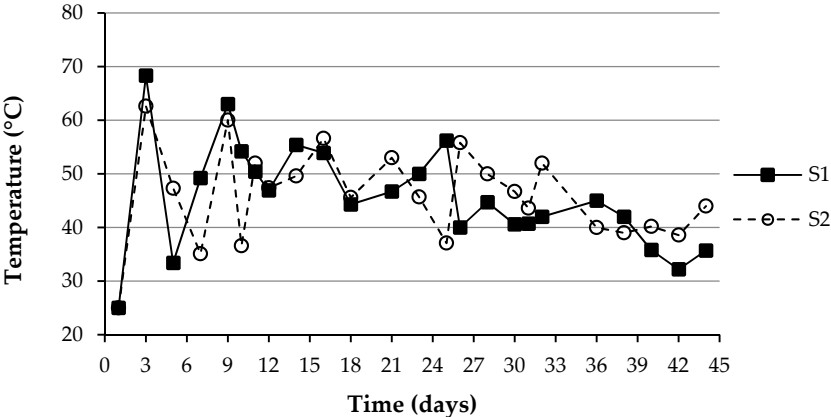

**Figure 3.** Progression of the temperature of the mixture during the composting process in S1 and S2 systems.

The moisture content of the product mixtures remained between 59 and 60% during the 44-day experiment in S1 and between 60 and 56% in S2 (Tables S1 and S2). These moisture contents are typical values when composting is conducted in portable closed aerobic composters. The slightly lower content in S2 can be attributed to the continuous low-flow air extraction. This airflow also affected the vs. content in the mixture, which—although it decreased with composting time in all treatments (Tables S1 and S2)—made the degradation rate lower in treatment S2 compared to S1 and the control. Chowdhury et al. [66] observed higher degradation rates in treatments with lower airflow, which allowed for better temperature maintenance.

The pH values increased with composting time in all treatments due to a progressive alkalinization of the medium, resulting from the loss of organic acids and the generation of ammonia by the decomposition of proteins [67]. The EC also tends to increase during composting due to the mineralization of organic matter, which leads to an increase in nutrient concentration. However, this was only observed in the case of S2. In the rest of the treatments, a decrease in EC was observed during the process, which could be due to leaching phenomena caused by excessive wetting of the mixture [68].

As the composting process progressed, the C/N ratio decreased from 12.1 to 11.9 in S1 and from 11.3 to 10.7 in S2. In S1, 8.7% of the C and 7.2% of the N present in the raw materials were lost, compared to 9.9% of the C and 4.8% of the N in S2. Carbon and nitrogen losses were due to the volatilization of $CO_2$ and $NH_3$ as a consequence of oxidative degradation of organic materials due to the activity of aerobic microorganisms. Other authors [66,69] have reported higher C and N losses, mainly due to a higher C/N ratio.

During composting, the volume of waste is reduced due to the transformation of organic matter by microorganisms, which assimilate and metabolize it. In addition, a lot of water vapor is produced due to the high temperatures of the thermophilic phase, which also contributes to the volume reduction. For this reason, the nutrients (P, K, Na, etc.) present in the mixture were more concentrated at the end of the experiment in all treatments (Tables S1 and S2).

### 3.2. Vibrational Analysis of the Product Mixtures by Infrared Spectroscopy

Figure 4 shows the ATR-FTIR spectra of the product mixtures for the control, S1, and S2 treatments at the beginning of the process and after 44 days of composting. This type of analysis was performed to identify changes in composition.

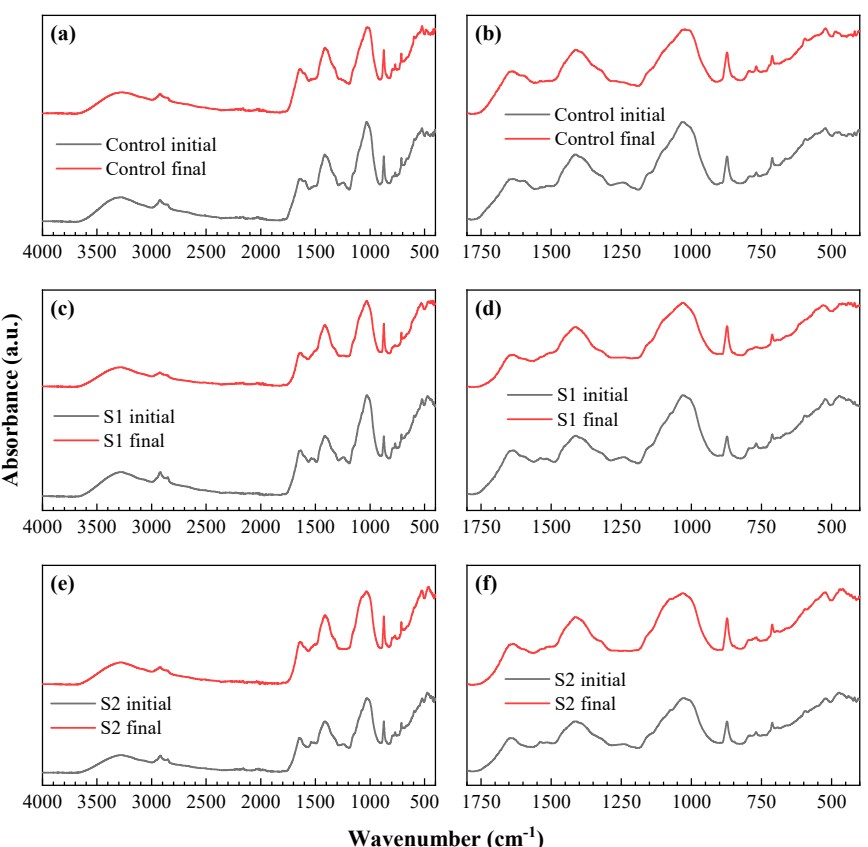

**Figure 4.** ATR−FTIR spectra of hen manure and wheat straw mixture samples at the beginning of the experiment and after 44 days for: (**a**,**b**) the control, (**c**,**d**) S1, and (**e**,**f**) S2 treatments. (**a**,**c**,**e**): spectra of each treatment over the whole 4000–500 cm$^{-1}$ range; (**b**,**d**,**f**): fingerprint region (1800–400 cm$^{-1}$ range).

The presence of a broad absorption band at 3279 cm$^{-1}$ in all spectra is attributable to O−H stretching vibration from the water contained in the product mixture samples, although a concurrence of the effect of N−H stretching vibration from amino compounds cannot be ruled out [70].

The bands at 2920 and 2850 cm$^{-1}$ are due to C−H stretching vibrations associated with $CH_2$ and $CH_3$ groups of aliphatic chains, although the absorption at 2920 cm$^{-1}$ has also been ascribed to asymmetric axial C−H deformation in methyl and methylene [71]. Around them, two weak bands are observed, whose origin is to be sought in the degradation of the hemicellulose produced by the microorganisms when contributing to their energetic function [72]. As the composting time progresses in both systems (S1 final, S2 final), the intensity of these bands decreases.

The pronounced band at 1634 cm$^{-1}$ can be attributed either to C=O of xylans (hemicellulose) or lignin, or to the C=C strain vibration (in olefins and possibly in aromatic compounds) [24]. It could also be ascribed to the asymmetric deformation vibration of ammonia coordinated to Lewis acid sites [73]. In any case, this absorption is a marker of the decomposition process leading to the formation of humic substances [74]. The bands at 2920/1634 cm$^{-1}$ decreased with increasing days of composting due to the biodegradation of organic biomass [75].

The band at 1412 cm$^{-1}$ is associated with the symmetric COO− stretching of carboxylates but is also due to the vibration of the aromatic rings of lignin. The increase in the intensity of this band with composting time (in both systems) indicates a predominance of aromatic carbons over aliphatic carbons, which could be related to higher stability and maturity of the product [76,77].

The small band at ca. 1250 cm$^{-1}$ is related to the presence of ammonia adsorbed on Lewis acid sites [73]. The intensity of this band is reduced in the 'S1 final' and 'S2 final' samples compared to the control, "S1 initial," and "S2 initial" samples. The fast-growing microorganisms produced a general uptake of nitrogen [78], which probably led to a decrease in NH$_3$ emissions as the composting process progressed.

The band at 1030 cm$^{-1}$ is attributable to C−O and C− stretching vibrations and the glycosidic bond's contribution (characteristic of polysaccharide structures). The intensity of this band has shown changes with the evolution of the composting process: as the composting time progresses, the intensity of the band increases in both systems ("S1 final" and "S2 final"), while the C/N ratio decreases, a sign of increased stability of the mixture. Torres-Climent et al. [79] also observed a negative correlation with the TOC/TN ratio.

The band at 871 cm$^{-1}$ denotes the presence of β (1→4) bonds between the xylose units in the hemicellulose, and the bands at 768 and 711 cm$^{-1}$ correspond to rocking and twisting modes, although some authors have referred them to the pyridine ring [80]. These bands are present in all treatments and do not differ in signal intensity.

### 3.3. Ammonia Emissions

Ammonia emissions (in ppm) inside each portable closed aerobic composter are shown in Figure 5. Ammonia emissions peaked in both systems on day 12 of the process, reaching 320 ppm in S1 and 210 ppm in S2, after which they decreased to 10 ppm on the last day. These results are consistent with those obtained by authors such as Sun et al. [56], who obtained a maximum concentration of NH$_3$ ranging from 500 to 600 ppm inside the membrane between days 1 and 10 of the composting period, or Chowdhury et al. [66], who found maximum emissions on the 6th and 11th day of the process. Previous studies have shown that NH$_3$ release in compost is mainly concentrated in the thermophilic period and gradually decreases in the maturity period [81].

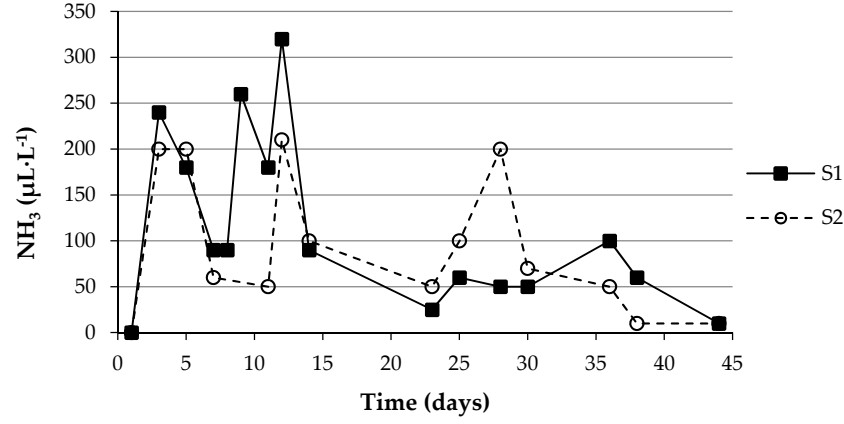

**Figure 5.** Progression of ammonia emissions in system 1 (S1) and system 2 (S2).

The high temperature reached in the thermophilic phase promoted the mineralization of organic N and increased the availability of ammonium nitrogen, which was the source of $NH_3$ emissions and, consequently, increased the loss of nitrogen in the mixture [64,66]. This process can be inferred from the increase in pH of the mixtures (Tables S1 and S2) as a result of the ammonification during the initial phase of the composting process. In later stages, nitrifying bacteria converted $NH_3$ to nitrate, so ammonium nitrogen started to decrease, which led to the rapid decrease of $NH_3$ concentration in the last stage [29,64].

The $NH_3$ emission rates in the S1 and S2 composting systems are presented in Figure 6.

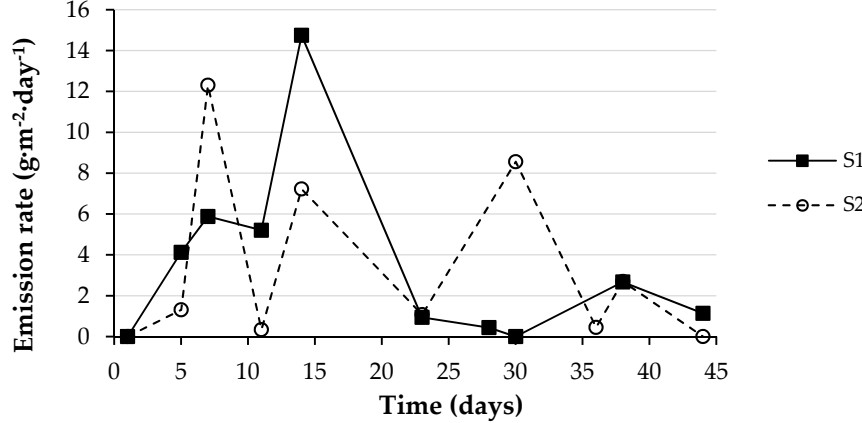

**Figure 6.** $NH_3$ emission rate in system 1 (S1) and system 2 (S2).

The $NH_3$ emission flux was highest in the early stages of the composting period in both systems, reaching a maximum value of 14.7 $g \cdot m^{-2} \cdot day^{-1}$ on day 14 for S1 and 12.3 $g \cdot m^{-2} \cdot day^{-1}$ on day 7 for S2, although in this case high values were also observed on days 14 and 30. The average emission rate was 3.5 and 3.4 $g \cdot m^{-2} \cdot day^{-1}$ for S1 and S2, respectively. Ammonia production during composting is related to the degradation of easily degradable organic substances, which is characteristic of the early stages of composting [56], so it is reasonable to obtain the highest emission rates in the first weeks.

The $NH_3$ emission flux in the S2 system remained higher—and over a longer period—compared to S1. As with the temperature, the possible continuous air inflow may favor the thermophilic stage to have a longer duration than in S1.

The emission rates achieved in this study were lower than those obtained by Sun et al. [56], who obtained a maximum $NH_3$ emission rate inside the membrane of up to 37.76 $g \cdot m^{-2} \cdot day^{-1}$, with an average emission rate of 8.8 $g \cdot m^{-2} \cdot day^{-1}$. This can be readily explained by the lower $NH_3$ concentrations measured in the air of the closed aerobic composters in this study.

Conversely, the emission rates obtained in this study were higher than those found by Fang et al. [82] in the composting of the solid fraction of dairy manure with a membrane cover (CT), for which maximum emission rates inside the membrane of 0.6–1.0 $g \cdot m^{-2} \cdot day^{-1}$ were registered in the first three days of the composting process. In their experiment, they observed that as the stack temperature increased, the condensation droplets formed under the membrane partially evaporated and $NH_3$ was discharged to the environment, thus decreasing the emission rate inside the membrane. In our study, the emission rate inside the compartment was not affected by $NH_3$ losses to the outside.

The accumulated $NH_3$ emissions during the 44 days of the composting process in the air chamber of the closed aerobic composting reactors of systems S1 and S2 are presented in Figure 7. The cumulative $NH_3$ emission from day 0 to day 14 in the S1 system was 1.7 times the value reached in S2. The difference was progressively reduced and, at the end of the experiment, on day 44, it was 33% higher (223 vs. 168 $g \cdot m^{-2}$). The values found are similar with small differences due to the heterogeneity of the compost mixture.

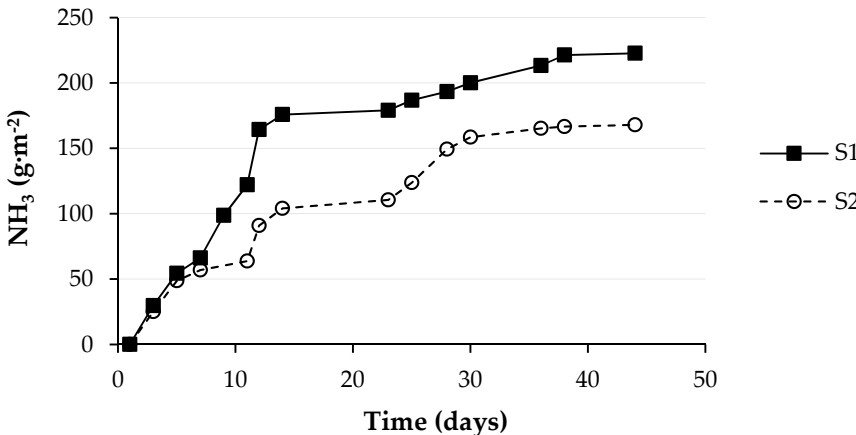

**Figure 7.** Cumulative $NH_3$ emission in the air chamber of the closed aerobic composting reactors of system 1 (S1) and system 2 (S2).

### 3.4. Operational Parameters of the GPM Systems

The experiments were carried out with continuous circulation of the acidic solution inside the membranes. The pH and EC values of the acidic solution are shown in Figure 8a,b for system S1 and system S2, respectively.

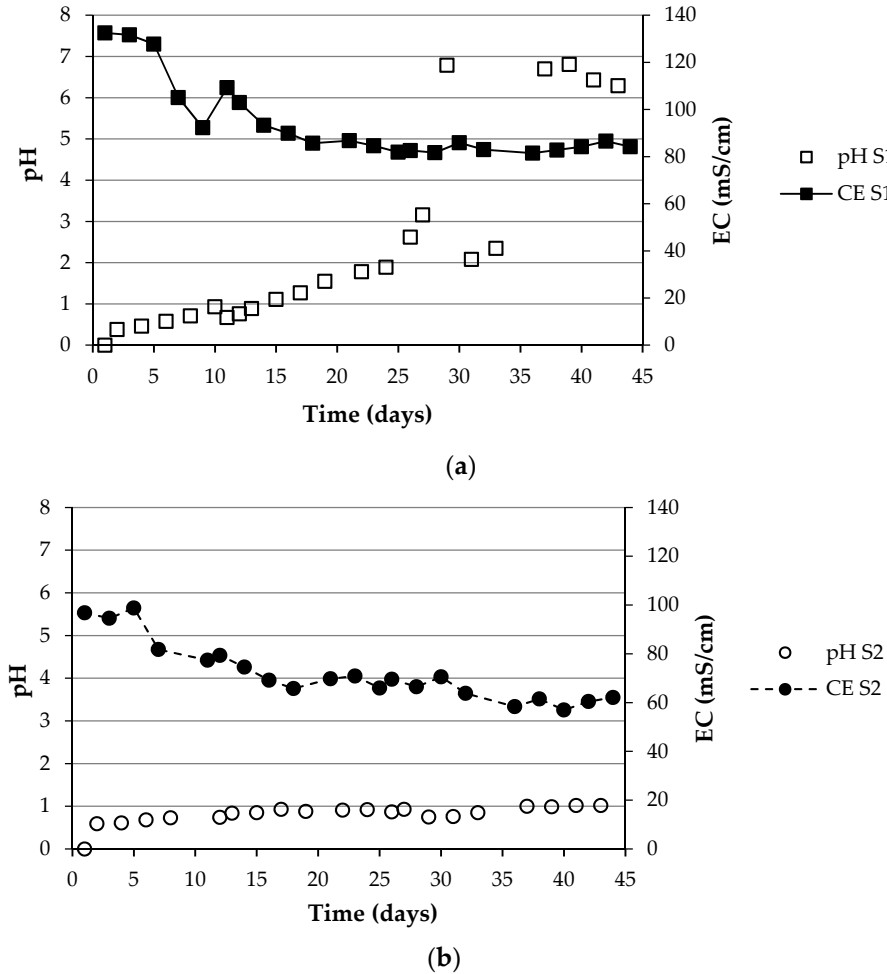

**Figure 8.** pH and electrical conductivity (EC) of the acidic solution in (**a**) system S1 and (**b**) system S2.

In system 1 (S1), a rapid increase in the pH of the acidic solution circulating inside the membranes was observed due to the increased $NH_3$ uptake (Figure 8a). The pH values above 2 in the acidic solution were corrected by the addition of concentrated $H_2SO_4$ in order to enhance $NH_3$ capture further, as indicated by other authors in their investigations on $NH_3$ capture from manure [83–86].

During the transfer of $NH_3$ across the membrane, a reduction of $H_3O^+$ occurs so that the alkalinity of the acidic solution increases (higher pH values). The EC values showed a direct relationship with the pH value and, in turn, with the concentration of TAN in the medium [87].

In system 2 (S2), a much slower increase of the pH of the acidic solution was observed (Figure 8b). The difference in pH change in both systems is due to two effects. First, $NH_3$ capture in membrane systems depends on the concentration gradient on both sides of the membrane. In the S2 system, the $NH_3$ generated in the composting process is extracted by the external prototype where the membranes are located. This leads to a lower $NH_3$ concentration and, therefore, to a slower $NH_3$ capture. Second, the volume of acidic solution in S2 was larger, so that a larger amount of $NH_3$ than in S1 would be needed for the pH change to be detected.

The TAN values (expressed in g) for systems S1 and S2 over the 44-day period are shown in Figure 9.

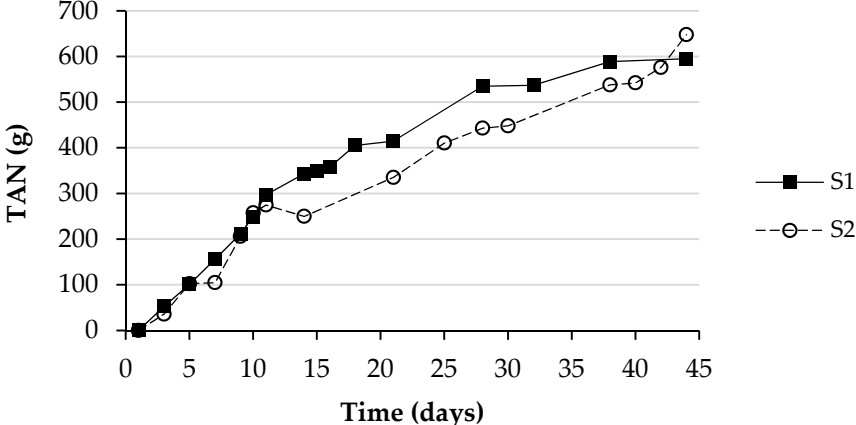

**Figure 9.** Total ammonia nitrogen (TAN) in the acidic solution for systems S1 and S2.

In both systems, a linear increase in the amount of TAN present in the acidic solution was observed at the beginning of the composting process and, from day 20 onwards, the capture remained at constant values. In the S1 system, concentrations close to 13,601 mg·$L^{-1}$ were reached and, in the S2 system, close to 4557 mg·$L^{-1}$. In absolute value, they correspond to 595 and 648 g TAN, respectively. The increase in capture is observed on the days with the highest $NH_3$ emission (Figure 5) and is also related to the pH and EC values of the acidic solution during the composting period (Figure 7).

To determine the effectiveness of the S1 and S2 systems in capturing the $NH_3$ released during the composting process, calculations were carried out to determine the TAN balance, as indicated in Section 2.5. The results are presented in Table 2.

**Table 2.** TAN balances in systems S1 and S2.

| Parameters | System 1 (S1) | | System 2 (S2) | |
|---|---|---|---|---|
| | Initial | Final | Initial | Final |
| Compost weight (kg) | 470 | 450 | 470 | 420 |
| Dry weight (‰) | 0.410 | 0.396 | 0.429 | 0.437 |
| TAN content (‰) | 0.029 | 0.027 | 0.032 | 0.030 |
| Total TAN content of compost (kg) | 5.65 | 4.85 | 6.35 | 5.51 |
| TAN emitted (kg) | 0.80 | | 0.85 | |
| TAN concentration in acidic solution (g·L$^{-1}$) | 13.6 | | 4.6 | |
| Volume of acidic solution (L) | 45 | 45 | 150 | 150 |
| Total TAN content in acidic solution (kg) | 0.61 | | 0.68 | |
| TAN recovery rate (g TAN·m$^{-2}$·day$^{-1}$) | 6.9 | | 1.9 | |

The nitrogen content in the manure was reduced by 14.2% and 13.3% for S1 and S2, respectively. Amounts of ammonia as TAN of 0.61 and 0.68 kg were recovered in acidic solution, with recovery rates of 6.9 and 1.9 g TAN·m$^{-2}$·day$^{-1}$ for S1 and S2, respectively. The $NH_3$ uptake was clearly influenced by the $NH_3$ concentration on the outside of the membrane. This is in agreement with the results of Rothrock et al. [86], who obtained $NH_3$ recovery rates of 28.63 and 10.42 g N·m$^{-2}$·day$^{-1}$ by applying 2% and 0% lime treatments, respectively, on poultry litter. They obtained a 12% higher $NH_3$ recovery in the 2% lime treatment and concluded that an important limiting factor for high $NH_3$ mass recovery is the $NH_3$ concentration in the air chamber of the experimental compartment. System 1 captures $NH_3$ directly from the closed aerobic composter at higher concentrations, which is why the capture efficiency was approximately six times higher. However, the amounts of $NH_3$ recovered by both systems were similar, probably due to the larger membrane surface area used in the S2 system. These results are in agreement with those obtained by Soto-Herranz et al. [55], who determined, at a laboratory scale—using synthetic solutions as a source of TAN—that approximately twice as much $NH_3$ was captured in suspended membrane systems when the membrane surface area was increased from 81.7 to 163.4 cm$^2$ and the TAN concentration in the emitting source was increased from 3000 to 6000 mg·L$^{-1}$.

The final product obtained in both systems was a stable concentrate of ammonium sulphate solution, with a maximum TAN concentration of 13.6 and 4.6 g TAN·L$^{-1}$ in S1 and S2, respectively, compared to an initial TAN content in the manure of 0.29 and 0.31 g TAN·L$^{-1}$, which is 15 to 47 times higher than in the manure.

Gas-permeable membrane technology thus can concentrate the $NH_3$ released during composting with several benefits for farmers: (1) greater control over nutrient application can be achieved, avoiding risks of N run-off or leaching; (2) reduced transport costs associated with manure application, as the reduced nitrogen concentration in the treated manure allows more manure to be spread closer to farms; and (3) marketing the $(NH_4)_2SO_4$ solution provides an extra source of income.

With 1.4 % N (13.6 g TAN·L$^{-1}$), the $(NH_4)_2SO_4$ solution obtained in this process could be considered for application in fertigation [88,89], with improvements in terms of plant uptake, yields and crop quality [88].

### 3.5. Comparison with Other GPM-Based Systems Used in Manure Management

After a comprehensive literature survey about studies focused on the application of GPM technology to manure management processes, no similar studies in which N is recovered from hen manure during composting have been found, thus preventing direct comparisons. Nonetheless, other relevant studies in which GPM technology has either been used for $NH_3$ recovery during manure management processes or in connection with composting processes (without N recovery) are discussed below. Examples of $NH_3$ recovery during composting processes using alternatives to GPM systems are also presented.

For instance, GPM technology has been successfully used in combination with anaerobic digestion (AD) to improve the process yield, limiting the inhibition of methanogenic

activity by high $NH_3$ concentrations. Chen et al. [90] employed vacuum-assisted submerged GPM technology (a PTFE hollow fiber membrane) on a laboratory scale to remove $NH_3$ produced during anaerobic digestion treatment of chicken manure, thus avoiding the problems caused by the inhibitory effect of $NH_3$. They were able to recover more than 80% of $NH_3$ in 6 h at 70 °C and 30 kPa, reducing the TAN concentration in the digestate and obtaining a 45% increase in $CH_4$ content as compared to the reactors without $NH_3$ removal. González-García et al. [91] studied the $NH_3$ removal effect applying submerged ePTFE GPM technology during an anaerobic digestion process of pig manure with sewage treatment plant anaerobic digester inoculum to improve the performance of the AD process. In the semicontinuous tests (205 days), they found that the TAN concentration in the digestate of the reactor with GPM system decreased by 23%, with a free $NH_3$ reduction of 54%. This resulted in an increase in methane production yield in the GPM reactor, which was 17% higher than in the other reactor, and in an 11% higher percentage of methane in the biogas.

Concerning composting, GPM systems have been used to reduce $NH_3$ emissions during the processes, albeit without N recovery. For example, Sun et al. [56] and Fang et al. [82] applied ePTFE-based GPM technology to pilot-scale open-top cow manure composting processes. In the work by Sun et al. [56], a 65% reduction in $NH_3$ emissions (comparing inside and outside the membrane) was attained after 48 days of composting, with the maximum $NH_3$ concentration reached outside the membrane being 58% lower than that present inside. Fang et al. [82], after 30 days of composting, obtained an 11.32% reduction in cumulative $NH_3$ emissions outside the membrane compared to inside. Thus, both studies concluded that GPM technology could effectively reduce $NH_3$ emissions during aerobic composting of cow manure.

Likewise, GPM technology has also been used to reduce emissions in closed composting systems on a laboratory scale. For instance, Ma et al. [92], during a 27-day composting experiment carried out with pig manure, observed a reduction of $NH_3$ emissions of 9.22% between the exterior and the interior of the membrane in the composting system. A higher reduction (30%) was observed for the system equipped with GPM technology as compared to a traditional aerobic composting system in the study by Sun et al. (2016) [93] during a 33-day pig manure composting process.

With reference to systems in which $NH_3$ has been recovered during composting processes, Kim et al. [94] used a condensation process (instead of GPM technology) to recover $NH_3$ produced during the composting of pig manure and sawdust in a closed system at a laboratory scale. Composting mixtures with different moisture percentages (55, 60, 65, and 70%) were processed over 56 days in reactors to which a condensation system was coupled. During the composting process, the liquid fractions were collected from closed systems through cooling, and the quantified ammonium nitrogen was recorded. For the optimized moisture-content (65%) compost mixture, they were able to recover up to 17,567 $mg \cdot L^{-1}$ on the second day, ca. 12,000 $mg \cdot L^{-1}$ on the third day, ca. 9000 $mg \cdot L^{-1}$ on the fourth day, and ca. 6000 $mg \cdot L^{-1}$ on the fifth day, after which the speed of volatilization (and capture) quickly decreased. No cumulative data were reported.

*3.6. Economic Assessment*

An economic assessment was carried out to estimate the potential viability of the two presented systems.

For the system based on gas-permeable membrane technology installed inside the closed aerobic composter (system S1), a summary of the costs for the modifications made to the standard closed aerobic composter is presented in Table S3, together with the approximate capital expenses, operating expenses, and operating revenues (Table S4). The values used in these calculations are based on the experimental data of this study and on the assumptions presented in Supplementary Materials.

Considering the volume of the composting unit (3.7 $m^3$) and the average N-$NH_3$ production (0.29 kg N $m^{-3}$), a total of 8.87 kg N-$NH_3 \cdot year^{-1}$ is produced (1.07 kg N-$NH_3$ are produced per batch). With a recovery efficiency of 76%, 6.74 kg N-$NH_3$ is recovered from

the composter each year. The maximum TAN recovery rate is 6.9 g TAN m$^{-2}$·day$^{-1}$ and a membrane surface of 1.96 m$^2$ is used. The annualized cost of equipment would account for 907 €·year$^{-1}$. In addition, the annual replacement of the membranes, considered at 20%, would amount to 61 €·year$^{-1}$. The chemical costs would be 4 €·year$^{-1}$ (0.29 €·kg$^{-1}$ of H$_2$SO$_4$) [95]. The energy costs for the composting process would be 626 €·year$^{-1}$, resulting from an average use of 15.7 kWh·day$^{-1}$ (unit cost in Spain = 0.1295 €·kWh$^{-1}$). The estimated annual cost for a composting system with a TAN recovery system installed inside the composting system would be 1598 €·year$^{-1}$. The sales of the compost produced would amount to 38 €·year$^{-1}$. The ammonium sulphate potentially recovered per year (6.74 kg N) has a fertilizer equivalent value of 16 €, assuming a value of 2.36 €·kg$^{-1}$ [95]. Therefore, the estimated net cost of the NH$_3$ recovery system is 1544 €·year$^{-1}$.

For system S2, in which the GPM system is installed in a separate compartment connected to the closed aerobic composter through a pipe, the estimated costs for the external compartment and the modifications made to the standard closed aerobic composter are presented in Table S5. The approximate capital expenses, operating expenses, and operating revenues are summarized in Table S6. The assumptions made are also presented in Supplementary Materials. Considering the volume of the composting unit (3.7 m$^3$) and the average N-NH$_3$ production (0.32 kg N·m$^{-3}$), a total of 9.79 kg N-NH$_3$ is produced (1.18 kg N-NH$_3$ is produced per batch). With a recovery efficiency of 80%, 7.83 kg N-NH$_3$ are recovered from the composter each year. The maximum TAN recovery rate is 1.9 g TAN·m$^{-2}$·day$^{-1}$, and a surface area of 7.7 m$^2$ of membrane is used. The initial cost of the external compartment and all the equipment associated with the operation of the GPM system, including the membranes, would account for 13,595 € (over twice that of system S1). The annual equipment costs would thus account for 2026 €·year$^{-1}$. In addition, the annual replacement of the membranes would account for 241 €·year$^{-1}$. The costs of the chemicals would be 10 €·year$^{-1}$. The energy costs for the composting process would be 626 €·year$^{-1}$, resulting from an average use of 15.7 kWh·day$^{-1}$. The estimated total annualized cost for the batch composting system with an external TAN recovery system would be 2903 €·year$^{-1}$. The sales of the compost produced would amount to 38 €·year$^{-1}$, and the ammonium sulphate potentially recovered per year (7.83 kg N) has a fertilizer equivalent value of 19 €. Therefore, the estimated net cost of ammonia recovery is 2846 €·year$^{-1}$ (84% higher than in system S1).

A comparison with other technologies is provided in Table 3, although a word of caution seems necessary, given that those studies recover NH$_3$ from the air of animal houses, not from a composting reactor.

**Table 3.** Comparison of net costs of NH$_3$ recovery from manure for different technologies.

| NH$_3$ Recovery Technology | NH$_3$ Source | Net Cost (€·Place$^{-1}$·Year$^{-1}$) | Reference |
|---|---|---|---|
| Biotrickling filter/biofilters | | 0.43 (broilers) 13.2 (pigs) | [96] |
| Acid scrubbing | Air from animal houses | 0.43 (broilers) 13.69 (pigs) | |
| Bioscrubbers | | 8.23–15.55 (pigs) | [97] |
| Air filtration | | 1.39 (pigs) | [98] |
| Air scrubbing | | 22–50 (sows) 4–15 (pigs) | [99] |
| GPM | Air from a closed aerobic composting reactor | 3.55 (free-range laying hen) | This work |

*3.7. Applicability of the Evaluated Membrane Composting Systems*

In view of the results obtained, the installation of system S1 would be effective in livestock facilities in which the volume of waste produced is not excessively high, or for

community composting, given that the implementation of membrane technology is still costly, 156.25 €·m$^{-2}$ (current market price). In this way, composting would be applied as a management technique for the waste produced, and the final product could be used as a growing medium or agricultural amendment. At the same time, the application of gas-permeable membrane technology during composting would allow the recovery of N released in the form of NH$_3$ during composting, producing an inorganic N-rich compost, reducing ammonia emissions to the atmosphere and thus contributing to the attainment of the emission control and reduction target set by Decision (EU) 2017/1757 [62], in which an ammonia emission ceiling of 353 kt is set. Further, it would be an alternative to the generation of conventional N fertilizers, whose production depends on natural gas and electricity, both of which are rising.

System S2 would be most useful in livestock facilities that have already applied some form of best available technology (BAT) for manure management to contribute to pollution prevention and control, such as solid–liquid separation or anaerobic digestion, in which case it would be interesting to couple gas-permeable membrane technology to these techniques to recover N and contribute to odor mitigation.

## 4. Conclusions

Gas-permeable membrane (GPM) technology has proved suitable for capturing NH$_3$ volatilized in aerobic composting processes, allowing the recovery of nitrogen lost in the form of NH$_3$ as a nonvolatile and concentrated ammonium salt that can be used as a fertilizer. NH$_3$ capture was found to be more efficient when the GPM system was installed inside the closed aerobic composting reactor (S1) than when it was installed in an external compartment connected to the reactor (S2), given that in the former configuration the membrane was in direct contact with a highly NH$_3$ concentrated atmosphere. The higher NH$_3$ concentration in the reactor's air chamber (up to 320 ppm) resulted in a TAN recovery rate of 6.9 g TAN·m$^{-2}$·day$^{-1}$ in system S1, 3.6 times higher than that attained in system S2 (1.9 g TAN·m$^{-2}$·day$^{-1}$). Based on these results, the installation of systems with S1 configuration is recommended in farms that generate low volumes of waste or for community composting, given that the cost of the membrane is the limiting factor for the application of this technology on a larger scale. The S2 configuration would be of particular interest as an alternative technology on farms that already have a waste treatment system, as it can easily be connected to an already available equipment.

**Supplementary Materials:** The following are available online at https://www.mdpi.com/article/10.3390/agronomy11122384/s1: Table S1: Physicochemical parameters of the initial product mixture samples in each treatment; Table S2: Physicochemical parameters of the product mixture samples at the end of the experiment (after 44 days) in each treatment; Table S3: Estimated costs for the modifications made to the standard closed aerobic composter in system S1; Table S4: Summary of capital expenses, operating expenses, and operating revenues for a closed aerobic composting system with an NH$_3$ capture system based on GPM technology directly installed in the aerobic composter; Table S5: Estimated costs for the external compartment and modifications made to the standard closed aerobic composter in system S2; Table S6: Summary of capital expenses, operating expenses, and operating revenues for a closed aerobic composting system connected to an external compartment with the NH$_3$ capture system based on GPM technology.

**Author Contributions:** Conceptualization, J.M.A.-R., M.S.-H. and M.S.-B.; data curation, M.S.-H.; formal analysis, M.S.-H., J.M.A.-R. and P.M.-R.; funding acquisition, M.S.-B.; investigation, M.S.-H. and J.M.A.-R.; methodology, M.S.-H. and J.M.A.-R.; resources, M.S.-B.; supervision, M.S.-B., J.M.A.-R. and P.M.-R.; validation, J.M.A.-R., M.S.-B. and P.M.-R.; visualization, M.S.-H. and P.M.-R.; writing—original draft, M.S.-H., M.S.-B., J.M.A.-R. and P.M.-R.; writing—review and editing, M.S.-H., J.M.A.-R. and P.M.-R. All authors have read and agreed to the published version of the manuscript.

**Funding:** This research was funded by the European Union, project LIFE+ Ammonia Trapping (LIFE15-ENV/ES/000284). The APC was funded by LIFE+ Ammonia Trapping (LIFE15-ENV/ES/000284) project.

**Institutional Review Board Statement:** Not applicable.

**Informed Consent Statement:** Not applicable.

**Data Availability Statement:** The data presented in this study are available on request from the corresponding author. The data are not publicly available due to their relevance as part of an ongoing Ph.D. thesis.

**Acknowledgments:** This work was funded by the European Union in the framework of the LIFE Project "Ammonia Trapping" (LIFE15-ENV/ES/000284) "Development of membrane devices to reduce ammonia emissions generated by manure in poultry and pig farms".

**Conflicts of Interest:** The authors declare no conflict of interest. The funders had no role in the design of the study; in the collection, analyses, or interpretation of data; in the writing of the manuscript, or in the decision to publish the results.

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
