# Peer review of "Reduction of Ammonia Emissions from Laying Hen Manure in a Closed Composting Process Using Gas-Permeable Membrane Technology"

_agronomy, doi:10.3390/agronomy11122384_

Round 1
Reviewer 1 Report
General comment:
This article investigated the applicability of two different GPM membrane systems to recover N released during the laying hen manure closed composting process. Here I have some questions. It confused me what is the current application status of GPM membrane system and why you chose the two different GPM membrane systems? The selection criteria of sulfuric acid concentration and flow rate? Generally, this paper has systematically studied the changes of physical and chemical indexes, ammonia emissions and ammonia recovery from laying hen manure in a closed composting process with two different GPM membrane systems, but it didn’t have some innovative description or investigation on the efficiency of ammonia recovery. The quality of the whole manuscript should be improved. Nonetheless, a large-scale experiment made good practical sense.
Other minor comments/suggestions are given to the authors:
- Abstract should be condensed. The raising of scientific question, a methodological part introduced and the results should be the response to the objectives and the idea presented on the title.
- Lines 74-75: “greatest economic” is too absolute. It should be replaced or add relevant references to prove it.
- Lines 91-95: It simply lists the literature and lacks a statement of relevance to your research.
- Introduction should explain the application, advantages and shortcomings of gas-permeable membranes systems.
- Introduction should clearly point out the innovation and scientific significance of this article.
- The reason for using the expanded polytetrafluoroethylene should be explained in the materials and methods. Why not use other materials?
- Lines 174-176: Why only analyze physical and chemical properties of the beginning and end of the 44-day period? I think analyzing the physical and chemical properties of all samples during composting is also important to explain ammonia emissions.
- Lines 203-206: Sample preparation method and original spectral treatment method should be added.
- Lines 228: “remained constant until day 8-10” is not accurate. There was a big drop in temperature after day 3?
- Lines 309: It is recommended that the spectra of each group be placed in one figure.
- Lines 328: CO2 emissions are not measured in this study, so this statement is not accurate. It is recommended to rewrite or add references.
- Lines 342-344: A comparative analysis of the ammonia cumulative emissions should be added.
- Lines 415: Can you draw your conclusions by comparing the two systems from the aspects of equipment input cost and energy consumption cost?
- I would suggest write more critically in the results and discussion section, I found it's written very superficial.
After careful reading of the whole manuscript, I feel that this paper presents information of interest to the readers of Agronomy but that significant revisions are needed prior to publication.
Author Response
Reviewer 1
General comment:
This article investigated the applicability of two different GPM membrane systems to recover N released during the laying hen manure closed composting process. Here I have some questions. It confused me what is the current application status of GPM membrane system and why you chose the two different GPM membrane systems? The selection criteria of sulfuric acid concentration and flow rate? Generally, this paper has systematically studied the changes of physical and chemical indexes, ammonia emissions and ammonia recovery from laying hen manure in a closed composting process with two different GPM membrane systems, but it didn’t have some innovative description or investigation on the efficiency of ammonia recovery. The quality of the whole manuscript should be improved. Nonetheless, a large-scale experiment made good practical sense.
Response: Regarding the first question raised by the Reviewer, GPM systems have been tested for NH3 recovery in anaerobic biodigesters, but not in closed aerobic composters. In relation to composting, membranes have been generally assayed to cover open piles, determining their potential to reduce emissions, but without any sort of NH3 recovery. A new subsection has been included in the discussion to comment on these works, describing the current application status of the technology. Further, a new paragraph has been included in the introduction to briefly comment on this status of the technology (minimizing the overlap with the new discussion section): “Authors such as Sun et al. and Ma et al. […] their promising results have encouraged the research presented herein.”
In connection with the second question on why two systems were studied, the last paragraph of the introduction has been re-written to clarify this point.
Concerning the choice of the capture solution (sulfuric acid) and its flow rate, please kindly note that these two factors (among others) were analyzed in a previous study [Effect of Acid Flow Rate, Membrane Surface Area, and Capture Solution on the Effectiveness of Suspended GPM Systems to Recover Ammonia. Membranes 2021, 11(7), 538; https://doi.org/10.3390/membranes11070538]. This point has now been clarified in subsection 2.1, to refer the interested reader to the article.
Other minor comments/suggestions are given to the authors:
- Abstract should be condensed. The raising of scientific question, a methodological part introduced and the results should be the response to the objectives and the idea presented on the title.
It has been corrected according to the Reviewer's indications and suggestions.
- Lines 74-75: “greatest economic” is too absolute. It should be replaced or add relevant references to prove it.
We agree with the Reviewer that the statement was not accurate and it has been deleted in the revised version: all listed emissions contribute to the loss of fertilizer value of compost.
- Lines 91-95: It simply lists the literature and lacks a statement of relevance to your research.
- Introduction should explain the application, advantages and shortcomings of gas-permeable membranes systems.
Comments 3 and 4 have been addressed by rewriting part of the introduction, which has been expanded with new information.
- Introduction should clearly point out the innovation and scientific significance of this article.
The paragraph on the aim of the paper has been rewritten to emphasize the scientific contribution of the study, explaining the current status of the technology and the research gap in the previous paragraph.
- The reason for using the expanded polytetrafluoroethylene should be explained in the materials and methods. Why not use other materials?
This question has been answered in the revised introduction section.
- Lines 174-176: Why only analyze physical and chemical properties of the beginning and end of the 44-day period? I think analyzing the physical and chemical properties of all samples during composting is also important to explain ammonia emissions.
We agree with the Reviewer that having information about the physico-chemical properties of all samples during composting would be interesting and would provide additional information about the composting process. However, our study was designed to focus specifically on the N balance and on estimating the NH3 capture efficiency, and we would like to keep that focus intact. Please note that a monitoring of the process was carried out during the 44 days, and that other data is provided for the intermediate steps (T, H, pH, EC and OM).
- Lines 203-206: Sample preparation method and original spectral treatment method should be added.
The corresponding materials and methods subsection has been updated to address the Reviewer's comment.
- Lines 228: “remained constant until day 8-10” is not accurate. There was a big drop in temperature after day 3?
The wording has been corrected and additional explanations have been added.
- Lines 309: It is recommended that the spectra of each group be placed in one figure.
The spectra have been grouped by treatment, as suggested by the Reviewer.
- Lines 328: CO2 emissions are not measured in this study, so this statement is not accurate. It is recommended to rewrite or add references.
The statement has been rewritten and an additional reference has been added.
- Lines 342-344: A comparative analysis of the ammonia cumulative emissions should be added.
A figure and a paragraph on cumulative ammonia emissions has been added to the revised results section, as requested.
- Lines 415: Can you draw your conclusions by comparing the two systems from the aspects of equipment input cost and energy consumption cost?
A new section entitled "Economic assessment" has been added to discuss the economic costs associated with the operation of the two aerobic composting systems equipped with the GPM systems.
- I would suggest write more critically in the results and discussion section, I found it's written very superficial.
The “Results and Discussion” section has been improved according to include additional comparisons with the literature.
After careful reading of the whole manuscript, I feel that this paper presents information of interest to the readers of Agronomy but that significant revisions are needed prior to publication.
Reviewer 2 Report
This article deals with the reduction of ammonia emission from laying hen mnaure using gas-permeable mebrane.
Ammonia emission, especially in EU, is one of the critical environment problem and it seems that this technology could be attrbuted to solve it.
There are some quenstions and comments.
line 40-57. Add and compare ammonia inventory content comparing with other country and/or other continent (also, spain's position in EU).
line 61. In addition, VOC and UV (for energy) are also important factors for PM formation.
line 86-95. Added more various techonolgy to reduce ammmonia emission (or ammonia trapping system in field scale).
Figure 1(c). For example, mark the solution tank correctly with a red circle.
What are the manufacturing costs of S1 and S2, respectively?
line 187. Assimilable means bioavailable fraction?
line 206 What is the difference between KBr and ZnSe?
line 223-234, Figure 3. The temperature means daily average? or the highest temperature every day?
Author Response
Reviewer 2
This article deals with the reduction of ammonia emission from laying hen manure using gas-permeable membrane.
Ammonia emission, especially in EU, is one of the critical environment problems and it seems that this technology could be attributed to solve it.
There are some questions and comments.
line 40-57. Add and compare ammonia inventory content comparing with other country and/or other continent (also, Spain's position in EU).
This information has been included in the introduction section, as requested by the Reviewer.
line 61. In addition, VOC and UV (for energy) are also important factors for PM formation.
We agree with the Reviewer's comment. Nonetheless, this study focuses on the emissions of ammonia, and other precursors of particulate matter are beyond the scope of the study.
line 86-95. Added more various technologies to reduce ammonia emission (or ammonia trapping system in field scale).
This issue has been addressed by expanding the introduction section.
Figure 1(c). For example, mark the solution tank correctly with a red circle.
The change suggested by the Reviewer has been made, and figure 1(c) has been updated accordingly.
What are the manufacturing costs of S1 and S2, respectively?
A new section "Economic Assessment" has been added to discuss the economic costs associated with the operation the two aerobic composting systems equipped with the GPM systems.
line 187. Assimilable means bioavailable fraction?
‘Assimilable phosphorus’ or ‘soluble phosphorus’ refers to plant-assimilable phosphorus or ‘available inorganic phosphorus’, which can include small amounts of organic phosphorus, as well as orthophosphate, the form taken up by plants. This has been clarified in the manuscript.
line 206 What is the difference between KBr and ZnSe?
Infrared spectroscopy incorporates several types of measurement methods. The classical methods are the KBr pellet method and the Nujol method (transmission mode), and require sample preparation. On the other hand, the Attenuated Total Reflection (ATR) method is able to measure powder samples directly. ATR method involves pressing the sample against a high-refractive-index prism and measuring the infrared spectrum using infrared light that is totally internally reflected in the prism. A zinc selenide (ZnSe) or germanium (Ge) prism is used in the ATR accessory. In comparison with the classical methods, ATR is an excellent method for obtaining infrared information for the powder sample surface. However, care is required with the wavenumber dependency of the absorption peak intensity (which has been corrected with Perkin-Elmer proprietary software in-built algorithm).
line 223-234, Figure 3. The temperature means daily average? or the highest temperature every day?
We refer to the average daily temperature. It has been clarified in the revised manuscript.

Reviewer 3 Report
Journal
Agronomy (ISSN 2073-4395)
Manuscript ID
agronomy-1464358
Type
Article
Title
Reduction of ammonia emissions from laying hen manure in a closed composting process using gas-permeable membrane technology
This study investigated the applicability of two GPM membrane systems to recover N released during the laying hen manure closed composting process. It is an interesting study, but it needs to be improved before further consideration.
Line 30 – reflect also the amount, which is irectly re-applied to soils as organic fertiliser. Some for other manure.
Extend information about f pathogenic bacteria.
Line 71-75: extend and provide references.
Do not use term “biodigester”, as it may lead towards confusion with biogas technology and anaerobic digestion. You can maybe apply term “aerobic biodigester”. Or some other appropriate alternative.
Work more on Discussion section, which should unambiguously express a comparison of the achieved results with the previous knowledge of the topic. It must make clear what is completely new in the presented results and where these results differ from the findings of other authors, and in what they coincide with the published opinions. Discussion should emphasise the significance of the results and draw attention to the newly opened issues and the need for their solution. Apply this comment to the whole Discussion section.
Example:
“The nitrogen content in the manure was reduced by 14.2% and 13.3% for S1 and S2, respectively. Amounts of ammonia as TAN of 0.61 and 0.65 kg were recovered in acidic solution, with recovery rates of 6.9 and 1.9 g TAN·m-2·day-1 or S1 and S2, respectively. The NH3 uptake was clearly influenced by the NH3 concentration on the outside of the embrane. This is in agreement with the results of Rothrock et al. [56], who obtained NH3 388 ecovery rates of 28.63 g N·m-2·day-1 and 10.42 g N·m-2·day-1 by applying 2% lime and 0% lime treatments, respectively, on poultry litter. They obtained a 12% higher NH3 recovery in the 2% lime treatment and concluded that an important limiting factor for high NH3 mass recovery was the NH3 concentration in the air chamber of the experimental compartment” I agree with all above stated, but in the discussion you need also to reflect – what is the actual meaning of it – what does it mean – what are the practical implications and so on.
What needs to be also improved is conclusion. Conclusion is written as summary now. It needs to be “Conclusion”, which is actually concluding. You need to sum up the new knowledge in contributions. The data presented in the results and discussion sections are not repeated in detail. The relationship between the achieved results and the aims of the research is shown. Try to conclude as well as provide recommendations. Currently, it is not saying much to us other than summary.
With best regards,
The Reviewer
Author Response
Reviewer 3
This study investigated the applicability of two GPM membrane systems to recover N released during the laying hen manure closed composting process. It is an interesting study, but it needs to be improved before further consideration.
Line 30 – reflect also the amount, which is directly re-applied to soils as organic fertiliser. Some for other manure.
We have not found exactly how much of the 1400 million tons of manure produced is applied to the soil. However, we have found the figure for the proportion of the manure generated that is treated, which is <10% of the manure produced. Therefore, it may be assumed that the difference is what is applied to the field.
Extend information about pathogenic bacteria.
Information on pathogenic bacteria present in poultry manure has been expanded.
Line 71-75: extend and provide references.
The information contained in the introductory section has been expanded to address the Reviewer's request.
Do not use term “biodigester”, as it may lead towards confusion with biogas technology and anaerobic digestion. You can maybe apply term “aerobic biodigester”. Or some other appropriate alternative.
The term biodigester has been replaced with "portable aerobic closed composter" or "closed aerobic composter" throughout the manuscript.
Work more on Discussion section, which should unambiguously express a comparison of the achieved results with the previous knowledge of the topic. It must make clear what is completely new in the presented results and where these results differ from the findings of other authors, and in what they coincide with the published opinions. Discussion should emphasize the significance of the results and draw attention to the newly opened issues and the need for their solution. Apply this comment to the whole Discussion section.
Example:
“The nitrogen content in the manure was reduced by 14.2% and 13.3% for S1 and S2, respectively. Amounts of ammonia as TAN of 0.61 and 0.65 kg were recovered in acidic solution, with recovery rates of 6.9 and 1.9 g TAN·m-2·day-1 or S1 and S2, respectively. The NH3 uptake was clearly influenced by the NH3 concentration on the outside of the embrane. This is in agreement with the results of Rothrock et al. [56], who obtained NH3 388 ecovery rates of 28.63 g N·m-2·day-1 and 10.42 g N·m-2·day-1 by applying 2% lime and 0% lime treatments, respectively, on poultry litter. They obtained a 12% higher NH3 recovery in the 2% lime treatment and concluded that an important limiting factor for high NH3 mass recovery was the NH3 concentration in the air chamber of the experimental compartment” I agree with all above stated, but in the discussion you need also to reflect – what is the actual meaning of it – what does it mean – what are the practical implications and so on.
The “Results and Discussion” section has been improved according to what has been found in the literature.
What needs to be also improved is conclusion. Conclusion is written as summary now. It needs to be “Conclusion”, which is actually concluding. You need to sum up the new knowledge in contributions. The data presented in the results and discussion sections are not repeated in detail. The relationship between the achieved results and the aims of the research is shown. Try to conclude as well as provide recommendations. Currently, it is not saying much to us other than summary.
The "Conclusions" section has been rewritten in line with the Reviewer's suggestion.
